

# Big data clustering techniques based on Spark: a literature review

Mozamel M. Saeed[1], Zaher Al Aghbari[2] and Mohammed Alsharidah[1]

[1] Department of Computer Science, Prince Sattam Bin Abdul Aziz, Riyadh, Saudi Arabia
[2] Department of Computer Science, University of Sharjah, Sharjah, United Arab Emirates

## ABSTRACT

A popular unsupervised learning method, known as clustering, is extensively used in data mining, machine learning and pattern recognition. The procedure involves grouping of single and distinct points in a group in such a way that they are either similar to each other or dissimilar to points of other clusters. Traditional clustering methods are greatly challenged by the recent massive growth of data. Therefore, several research works proposed novel designs for clustering methods that leverage the benefits of Big Data platforms, such as Apache Spark, which is designed for fast and distributed massive data processing. However, Spark-based clustering research is still in its early days. In this systematic survey, we investigate the existing Spark-based clustering methods in terms of their support to the characteristics Big Data. Moreover, we propose a new taxonomy for the Spark-based clustering methods. To the best of our knowledge, no survey has been conducted on Spark-based clustering of Big Data. Therefore, this survey aims to present a comprehensive summary of the previous studies in the field of Big Data clustering using Apache Spark during the span of 2010–2020. This survey also highlights the new research directions in the field of clustering massive data.

## INTRODUCTION

With the emergence of 5G technologies, a tremendous amount of data is being generated very quickly, which turns into a massive amount that is termed as Big Data. The attributes of Big Data such as huge volume, a diverse variety of data, high velocity and multivalued data make data analytics difficult. Moreover, extracting meaningful information from such volumes of data is not an easy task (*Bhadani & Jothimani, 2016*). As an indispensable tool of data mining, clustering algorithms play an essential role in big data analysis. Clustering methods are mainly divided into density-based, partition-based, hierarchical, and model-based clustering.

All these clustering methods are developed to tackle the same problems of grouping single and distinct points in a group in such a way that they are either similar to each other or dissimilar to points of other clusters. They work as follows: (1) randomly select initial clusters and (2) iteratively optimize the clusters until an optimal solution is reached (*Dave & Gianey, 2016*). Clustering has an enormous application. For instance, clustering is used in intrusion detection system for the detection of anomaly behaviours (*Othman et al., 2018*; *Hu et al., 2018*). Clustering is also used extensively in text analysis to classify documents

Corresponding author
Mozamel M. Saeed,
m.musa@psau.edu.sa,
mozamel8888@gmail.com

into different categories (*Fasheng & Xiong, 2011*; *Baltas, Kanavos & Tsakalidis, 0000*). However, as the scale of the data generated by modern technologies is rising exponentially, these methods become computationally expensive and do not scale up to very large datasets. Thus, they are unable to meet the current demand of contemporary data-intensive applications (*Ajin & Kumar, 2016*). To handle big data, clustering algorithms must be able to extract patterns from data that are unstructured, massive and heterogeneous.

Apache Spark is an open-source platform designed for fast-distributed big data processing. Primarily, Spark refers to a parallel computing architecture that offers several advanced services such machine learning algorithms and real time stream processing (*Shoro & Soomro, 2015*). As such, Spark is gaining new momentum, a trend that has seen the onset of wide adoption by enterprises because of its relative advantages. Spark grabbed the attention of researchers for processing big data because of its supremacy over other frameworks like Hadoop MapReduce (*Verma, Mansuri & Jain, 2016*). Spark can also run in Hadoop clusters and access any Hadoop data source. Moreover, Spark Parallelization of clustering algorithms is an active research problem, and researchers are finding ways for improving the performance of clustering algorithms. The implementation of clustering algorithms using spark has recently attracted a lot of research interests.

This survey presents the state-of-the-art research on clustering algorithms using Spark Platform. Research on this topic is relatively new. Efforts started to increase in the last few years, after the Big Data platform, such as Apache Spark, was developed. This resulted in a number of research works that designed clustering algorithms to take advantage of the Big Data platforms, especially Spark due to its speed advantage. Therefore, review articles are needed to show an overview of the methodologies of clustering Big Data, highlight the findings of research and find the existing gaps this area.

Consequently, few researchers have published review articles (*Labrinidis & Jagadish, 2012*; *Gousios, 2018*; *Aziz, Zaidouni & Bellafkih, 0000*; *Salloum et al., 2016*; *Mishra, Pathan & Murthy, 2018*; *Aziz, Zaidouni & Bellafkih, 2018*; *Assefi et al., 2017*; *Armbrust et al., 2015*; *Xin et al., 2013*; *Xu & Tian, 2015*; *Zerhari, Lahcen & Mouline, 2015*). These review articles are either before 2016 or do not present a comprehensive discussion on all types of clustering methods. Therefore, a comprehensive review on clustering algorithms of big data using Apache Spark is needed because it is conducted based on a scientific search strategy. To the best of our knowledge, no survey has been conducted on Spark-based clustering of Big Data. For this purpose, this survey aims to present a comprehensive summary of the previous studies in the field of Big Data clustering using Apache Spark during the span of 2010–2020. The contributions of this review are:

- This review includes quality literature from pre-defined resources and based on pre-defined inclusion/exclusion criteria. Therefore, out of the 476 full-text articles studied, 91 articles were included.
- A taxonomy of Spark-based clustering methods that may point researchers to new techniques or new research areas.

- A comprehensive discussion on the existing Spark-based clustering methods and the research gaps in this area. Furthermore, we presented some suggestions for new research directions.

We believe that researchers in the general area of cluster Big Data and specially those designing and developing Spark-based clustering would benefit from the findings of this comprehensive review.

The rest of this survey is organised as follows. 'Background' presents the related surveys to the topic of clustering Big data. In 'Literature Review', we present a background on the Apache Spark. 'Survey Methodology' explains the methodology used in this survey. 'Survey Methodology' discusses the different Spark clustering algorithms. In 'Discussion and Future Direction', we present our discussion the clustering big data using Spark and future work. Finally, we conclude the paper in 'Conclusions'.

## BACKGROUND

Over the last decade, a huge amount of data has been generated. This increase in data volume is attributed to the growing adoption of mobile phones, cloud-based applications, artificial Intelligence and Internet of Things. Contemporary data come from different sources with high volume, variety and velocity, which make the process of mining extremely challenging and time consuming (*Labrinidis & Jagadish, 2012*). These factors have motivated the academic and the industrial communities to develop various distributed frameworks to handle the complexity of modern datasets in a reasonable amount of time.

In this regard, Apache spark, a cluster computing, is an emerging parallel platform that is cost-effective, fast, fault-tolerant and scalable. Thereby, such features make Spark an ideal platform for the dynamic nature of the contemporary applications. Spark is designed to support a wide range of workloads including batch applications, iterative algorithms, interactive queries, and streaming (*Gousios, 2018*). Spark extends the Hadoop model and support features such as in-memory computation and resilient distributed dataset, which make it significantly faster than the traditional Hadoop map-reduce for processing large data (*Aziz, Zaidouni & Bellafkih, 0000*).

As shown in Fig. 1, at the fundamental level, spark consist of two main components; A driver which takes the user code and convert it into multiple tasks which can be distributed across the hosts, and executors to perform the required tasks in parallel. Spark is based on RDD, which is a database tables that is distributed across the nodes of the cluster. Spark supports two main operations; Transformations; and actions. Transformation preform operations on the RDD and generates new one; Action operations are performed on RDD to produce the output (*Salloum et al., 2016*).

### Spark components
#### *Spark core*
Spark core is the foundation of Apache Spark and contains important functionalities, including components for task scheduling, memory management, fault recovery, interacting with storage systems. Spark Core is also home to the API that defines resilient

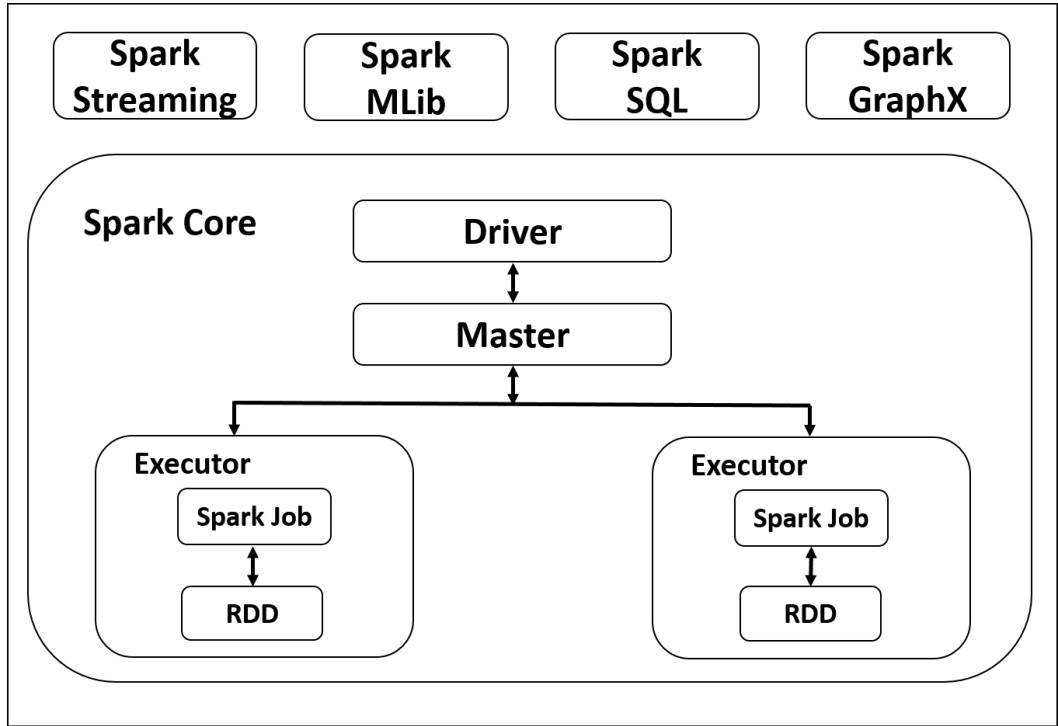

**Figure 1** **Apache Spark Architecture.**

distributed datasets (RDDs), which are Spark's main programming abstraction. RDDs represent a collection of items distributed across many compute nodes that can be manipulated in parallel. Spark Core provides many APIs for building and manipulating these collections (*Mishra, Pathan & Murthy, 2018*).

### Spark streaming

Spark streaming component provide scalable, high throughput API for processing of real-time stream data from various sources. Examples of data streams include logfiles generated by production web servers, or queues of messages containing status updates of a particular system (*Aziz, Zaidouni & Bellafkih, 2018*).

### Spark MLlib

Spark comes with a library MLlib which supports several common Machine Learning algorithms that include classification, regression, clustering, features extraction, transformation and dimensionality reductions (*Assefi et al., 2017*).

### Spark SQL

Spark SQL (*Armbrust et al., 2015*) is a module for processing structured data, which also enables users to perform SQL queries. This module is based on the RDD abstraction by providing Spark core engine with more information about the structure of the data.

### *Spark graphx*

GraphX (*Xin et al., 2013*) is a library for manipulating graphs (e.g., a social network's friend graph) and performing graph-parallel computations. Like Spark Streaming and Spark SQL, GraphX extends the Spark RDD API, allowing us to create a directed graph with arbitrary properties attached to each vertex and edge. GraphX also provides various operators for manipulating graphs (e.g., subgraph and mapVertices) and a library of common graph algorithms.

## Clustering big data

Clustering is a popular unsupervised method and an essential tool for Big Data Analysis. Clustering can be used either as a pre-processing step to reduce data dimensionality before running the learning algorithm, or as a statistical tool to discover useful patterns within a dataset. Clustering methods are based on iterative optimization (*Xu & Tian, 2015*). Although these methods are effective in extracting useful pattern from datasets, they consume massive computing resources and come with high computational costs due to the high dimensionality associated with contemporary data applications (*Zerhari, Lahcen & Mouline, 2015*).

## Challenges of clustering big data

The challenges of clustering big data are characterized into three main components:

1. **Volume:** as the scale of the data generated by modern technologies is rising exponentially, clustering methods become computationally expensive and do not scale up to very large datasets.
2. **Velocity:** this refers to the rate of speed in which data is incoming to the system. Dealing with high velocity data requires the development of more dynamic clustering methods to derive useful information in real time.
3. **Variety:** Current data are heterogeneous and mostly unstructured, which make the issue to manage, merge and govern data extremely challenging.

Conventional clustering algorithms cannot handle the complexity of big data due the above reasons. For example, k-means algorithm is an NP-hard, even when the number of clusters is small. Consequently, scalability is a major challenge in big data. Traditional clustering methods were developed to run over a single machine and various techniques are used to improve their performance. For instance, sampling method is used to perform clustering on samples of the data and then generalize it to the whole dataset. This reduces the amount of memory needed to process the data but results in lower accuracy. Another technique is features reduction where the dimension of the dataset is projected into lower dimensional space to speed up the process of mining (*Shirkhorshidi et al., 0000*). Nevertheless, the constant growth in big data volume exceeds the capacity of a single machine, which underline the need for clustering algorithms that can run in parallel across multiple machines. For this purpose, Apache spark has been widely adapted to cope with big data clustering issues. Spark provides in-memory, distributed and iterative computation, which is particularly useful for performing clustering computation. It also provides advanced local data caching system, fault-tolerant mechanism and faster-distributed file system.

## LITERATURE REVIEW

The topic of clustering big data using Spark platform have not been adequately investigated by academia. This suggests a comprehensive survey on research works in this regard. The literature in this area has already come up with some surveys and taxonomies, but most of them are related to Hadoop platform while others are outdated or do not cover every aspect of clustering big data using Spark.

The work in *Rotsnarani & Mrutyunjaya (2015)* conducted a survey on Hadoop framework for big data processing. Different features of Hadoop map-reduce are discussed to deal with the problems of scalability and complexity for processing big data. *Rujal & Dabhi (2016)* conducted a survey on k-means using map reduce model. In this article the technical details of parallelizing k-means using Apache Hadoop is discussed. According to this research, k-means method is regarded as a viable approach for certain applications of big data clustering and has attracted many researchers than any other techniques. On the other hand, (*Sood & Singh, 2019*) conducted a survey on the major challenges for big data processing using Hadoop map- reduce. According to this survey, network latency is the main limitation of Hadoop.

The authors of *Manwal & Gupta (2017)* conducted a survey on big data and Hadoop architecture. The paper classifies existing Hadoop based systems and discusses their advantages and disadvantages. The paper explains the different technologies (Hbase, Hive, Pig, etc.) used with the Hadoop distributed file system (HDFS). *Jiang et al. (2010)* conducted a survey on large scale data processing using Hadoop over the cloud. The main components of Hadoop platform and their functionalities are discussed.

The work in *Shanjiang et al. (2018)* conducted a comprehensive survey on spark ecosystem for processing large-scale data. In this article, spark architecture and programming model is introduced. The authors discussed the pros and cos of spark platform as well as the various optimization techniques used for improving spark performance for processing large scale data.

In *Maheshwar & Haritha (2016)*, the authors discussed the advantages of spark over the Hadoop map-reduce model. *Huang et al. (2017)* conducted a survey on the parallelization of density-based clustering algorithm for spatial data mining based on spark. The authors of *Ketu & Agarwal (2015)* conducted a performance evaluation of k-means over spark and map- reduce. On the other hand, a performance evaluation of three versions of k-means clustering for biomedical data using spark was conducted in *Shobanadevi & Maragatham (2017)*. A performance evaluation of parallel k-means with optimization algorithms for clustering big data using spark was conducted in *Santhi & Jose (2018)*. However, all the above surveys are either before 2016 or do not present a comprehensive discussion on all types of clusters. Therefore, a comprehensive survey on clustering algorithms of big data using Apache Spark is required to assess the current state-of-the-art and outline the future directions of clustering big data.

## SURVEY METHODOLOGY

The subject matter reviewed in this article is based on a literature review in clustering methods using Apache spark. We searched for the works regarding this topic and classify them into different Clustering techniques. All these papers talk about optimizing clustering techniques to solve the issues of big data clustering problems for various problems, viz., improve clustering accuracy, minimize execution time, increase throughput and scalability. Particularly, we are addressing the following questions:

- What are the types of Spark-based clustering methods?
- Which methods were used in the literature to cluster Big Data?
- What are the gaps in this research area?
- What optimization techniques were used in clustering?
- What are the pros and cons of the different Spark-based clustering methods?

### Search strategy

To narrow the scope of the searching for relevant papers to be included in this study, we used the ''AND'' and ''OR'' Boolean operators to combine the terms related to Spark-based clustering of Big Data. The following terms are used to find the relevant papers.

- ''Clustering big data using spark'',
- ''Apache Spark for Big data'',
- ''Clustering Big Data'',
- ''Clustering methods'',
- ''Data partitioning'',
- ''Big Data Partitioning'',
- ''Data segmentation''.

The papers relevant to Spark-based clustering of Big Data were retrieved from the following online sources.

- IEEE Explorer
- Springer,
- Elsevier
- ScienceDirect,
- Google Scholar,
- Researchgate,

### Paper filtering

Initially 1,230 and additional 43 reference books papers were identified through our search using the previously explained research strategies. As shown in Fig. 2, 797 of these were eliminated via our exclusion criteria. 476 papers were remaining. By reading and analysing the full-text articles, 385 of them were excluded. Irrelevant papers were removed by applying the exclusion criteria (shown below). In addition, duplicate papers retrieved from multiple sources were removed. Finally, 91 articles were included in this survey. The following inclusion/exclusion rules are applied on these papers.

- Inclusion criteria:

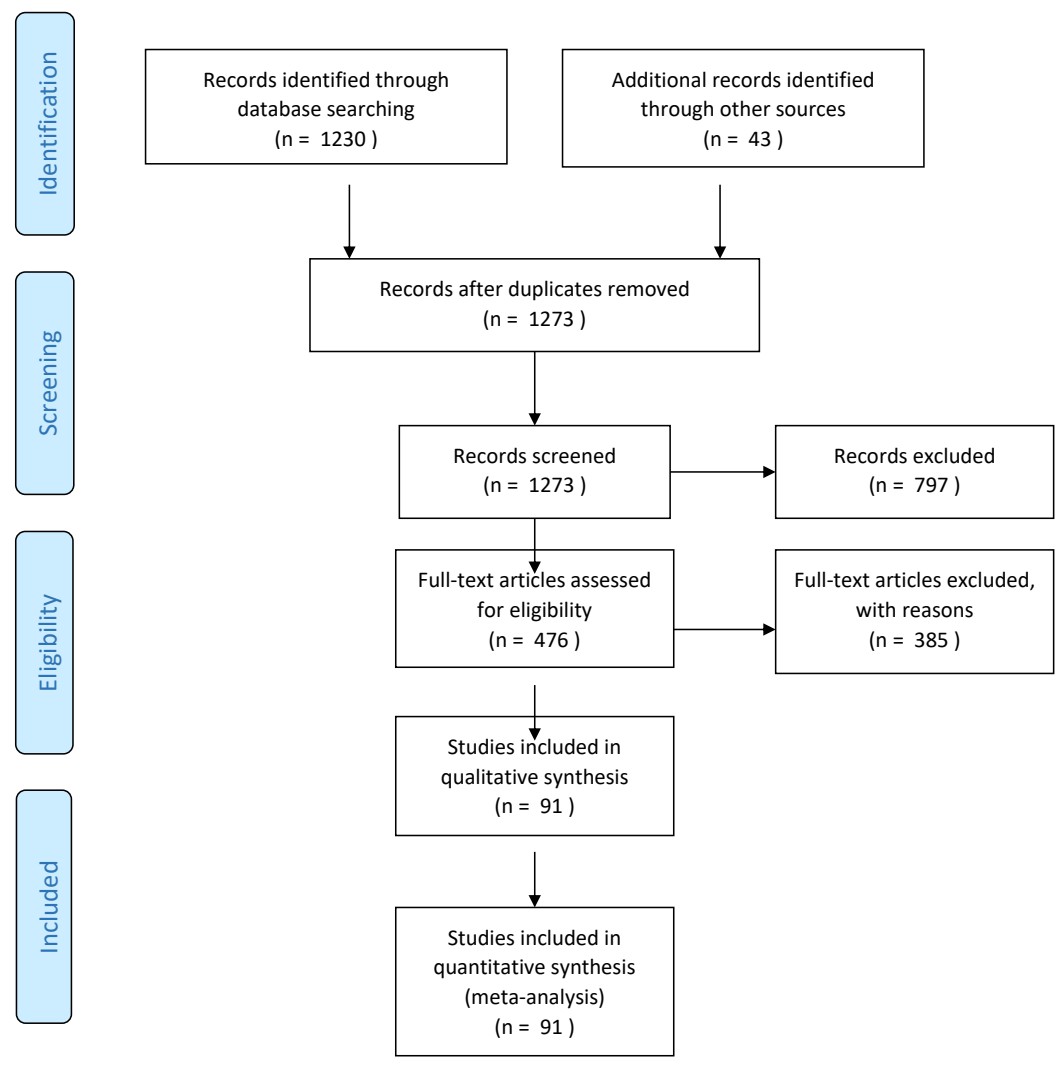

**Figure 2  Flowchart for paper exclusion.**

- – papers published within the period from January 2010 to April 2020.
- – papers in the area of Spark-based Big data clustering.
- – papers written in English language.

- Exclusion criteria:

  - – papers on clustering but not on Big data.
  - – papers that are not using a Big data platform such as Spark. papers with no clear publication information, such as publisher, year, etc.

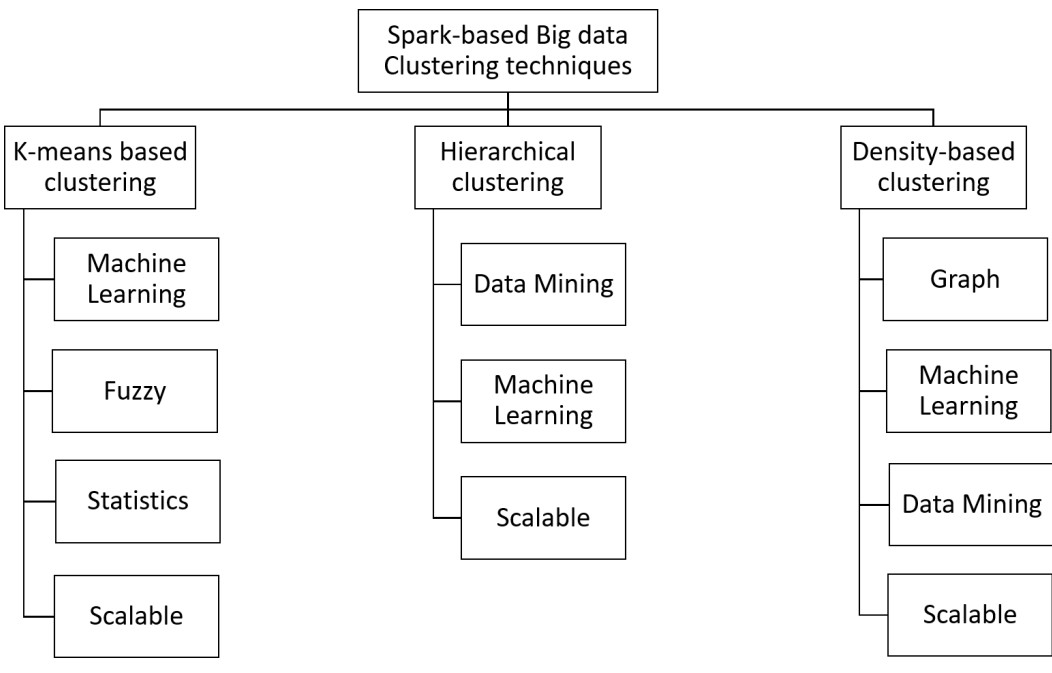

**Figure 3  Taxonomy of Spark-based clustering methods.**

## Spark-based clustering algorithms

In this work, the taxonomy of Spark-based Big Data clustering is developed to cover all the existing methods. Figure 3 shows the developed taxonomy.

**Survey Findings:** The research questions (see 'Survey Methodology') that we investigated in this survey are addressed as shown below:

- Answer to Q1: The Spark-based clustering algorithms were divided into three main categories: k-means based methods, hierarchal-based methods and density based methods. Each of these main categories were divided further into subcategories as depicted in Fig. 3. A detailed discussion of the Spark-based clustering methods in these subcategories is presented in the subsection below 'k-means based Clustering', 'Hierarchical clustering' and 'Density based-clustering' Fig. 3 and Table 1).
- Answer to Q2: We discuss the different methods that have been proposed in the literature under each of the three main Spark-based clustering categories in 'k-means based Clustering', 'Hierarchical clustering' and 'Density based-clustering'. The methods in these subsections are grouped based on their similarities in the approach. This grouping of the discussed methods is shown in Table 1.
- Answer to Q3: The gaps in the Spark-based clustering field are identified into two main points. The first is the lack of utilizing AI tools in clustering data and lack of using Big Data platforms. The second is that most current clustering methods do not support the characteristics of variety and velocity of Big Data. More discussion on this issue is in 'Discussion and Future Direction'.

**Table 1** Comparison of Spark-based Clustering methods in terms of the supported Big Data characteristic (volume, variety and velocity) and in terms of the type of data (real and synthetic) the proposed method was validated.

| Category | Sub-category | Paper | Supported Big Data Characteristic | | | Validated on | |
|---|---|---|---|---|---|---|---|
| | | | volume | variety | velocity | Real | Synthetic |
| K-means | Machine Learning | Kusuma et al. (2016) | ✓ | | ✓ | | ✓ |
| | | Sarazin, Lebbah & Azzag (2014) | ✓ | | | | ✓ |
| | | Gao & Zhang (2017) | ✓ | | | ✓ | |
| | | Thakur & Dharavath (2018) | ✓ | | | | ✓ |
| | | Lighari & Hussain (2017) | ✓ | | | ✓ | |
| | | Kamaruddin, Ravi & Mayank (0000) | ✓ | | ✓ | ✓ | ✓ |
| | Fuzzy | Wu et al. (2017) | | ✓ | | ✓ | |
| | | Win et al. (2019a) | ✓ | | | ✓ | |
| | | Win et al. (2019b) | ✓ | | | ✓ | |
| | | Liu et al. (2019) | ✓ | | | ✓ | |
| | | Bharill, Tiwari & Malviya (0000) | ✓ | | | ✓ | |
| | | Shah (2016) | ✓ | | | ✓ | |
| | Statistics | Lavanya, Sairabanu & Jain (2019) | ✓ | | | ✓ | |
| | | Pang et al. (0000) | ✓ | | | ✓ | |
| | | Chakravorty et al. (2014) | ✓ | ✓ | ✓ | ✓ | |
| | | Wang et al. (2016) | ✓ | ✓ | | ✓ | |
| | | Sinha & Jana (2016) | ✓ | | | | ✓ |
| | | Backhoff & Ntoutsi (2016) | | ✓ | ✓ | ✓ | |
| | | Ding et al. (2017) | ✓ | | | ✓ | |
| | | Sharma, Shokeen & Mathur (2016) | ✓ | | | ✓ | |
| | Scalable | Ben HajKacem, Ben N'Cir & Essoussi (2017) | ✓ | ✓ | | ✓ | ✓ |
| | | Ben HajKacem, Ben N'cir & Essoussi (0000) | ✓ | ✓ | | ✓ | ✓ |
| | | Zayani, Ben N'Cir & Essoussi (2016) | ✓ | | | ✓ | ✓ |
| | | Chitrakar & Petrovic (2018) | ✓ | | | ✓ | |
| | | Fatta & Al Ghamdi (2019) | ✓ | | | ✓ | |
| | | Zhang et al. (2019) | ✓ | | | ✓ | |
| | | Solaimani et al. (2014) | ✓ | ✓ | ✓ | ✓ | |
| | | Mallios et al. (0000) | ✓ | | | ✓ | |
| | | Guo, Zhang & Zhang (2016) | ✓ | | | ✓ | |
| | Data Mining | Ianni et al. (2020) | ✓ | | | | ✓ |
| | | Lee & Kim (2018) | ✓ | | | | ✓ |
| Hierarchical | Machine Learning | Sarazin, Azzag & Lebbah (2014) | ✓ | | | ✓ | ✓ |
| | | Malondkar et al. (2019) | ✓ | ✓ | ✓ | ✓ | |
| | Scalable | Jin et al. (2015) | ✓ | | | ✓ | |
| | | Solaimani et al. (0000) | ✓ | ✓ | ✓ | ✓ | |
| | | Hassani et al. (2016) | | | | | ✓ |

**Table 1** (*continued*)

| Category | Sub-category | Paper | Supported Big Data Characteristic | | | Validated on | |
|---|---|---|---|---|---|---|---|
| | | | **volume** | **variety** | **velocity** | **Real** | **Synthetic** |
| Density | Graph | *Rui et al. (2017)* | ✓ | | | ✓ | |
| | | *Zhou & Wang (0000)* | ✓ | | | ✓ | ✓ |
| | | *Kim et al. (2018)* | ✓ | | | ✓ | |
| | | *Lulli, Dell'Amico & Ricci (2016)* | ✓ | | | ✓ | ✓ |
| | Data Mining | *Han et al. (2018a)* | ✓ | | | | ✓ |
| | | *Hosseini & Kourosh (2019)* | ✓ | | | ✓ | ✓ |
| | | *Aryal & Wang (2018)* | ✓ | | | ✓ | |
| | Machine Learning | *Hosseini & Kiani (2018)* | ✓ | ✓ | | ✓ | |
| | | *Corizzo et al. (2019)* | ✓ | ✓ | | ✓ | ✓ |
| | | *Liang et al. (2017)* | ✓ | | | ✓ | ✓ |
| | | *Luo et al. (2016)* | ✓ | | | ✓ | |
| | Scalable | *Han et al. (2018b)* | ✓ | | | ✓ | ✓ |
| | | *Baralis, Garza & Pastor (2018)* | ✓ | | | | ✓ |
| | | *Gong, Sinnott & Rimba (0000)* | | | ✓ | ✓ | |

- Answer to Q4: Some existing works employed optimization techniques to improve clustering results. These optimization techniques were mainly used with k-means methods as discussed in 'Fuzzy based Methods' and 'Clustering Optimization'.
- Answer to Q5: The pros and cons of the different methods are discussed in the 'k-means based Clustering', 'Hierarchical clustering' and 'Density based-clustering', that discuss the different types of Spark-based clustering methods. We also discuss our findings related to the Spark-based clustering methods in 'Discussion and Future Direction'.

## k-means based clustering

This method divides the data into disjoint clusters of similar points. In each cluster, a central point is obtained via a distance function and is considered as the centroid of all other points within the cluster. The clusters are iteratively optimized until an optimal solution is reached.

k-mean is a framework of clustering or a family of distance functions, which provides the basis for different variants of k-mean algorithms. k means is extensively used in clustering big data due to its simplicity and fast convergence. One major backward of k-means is the priori setting of the number of clusters, which have significant effect on the accuracy of final classification (*Hartigan, Wong & Algorithm, 1979*). In addition, k-means is not suited in situations where the clusters do not show convex distributed or vary in sizes (*Jain, 2010*). Due to these limitations, several modifications of k -means have been proposed such as fuzzy k-means and k-means++ (*Huang, 1998*). Several works have been conducted to execute k-means effectively under the Spark framework to improve its performance and scalability. Therefore, the Spark-based k-means methods can be divided into four subcategories: Machine Learning based methods, Fuzzy based methods, Statistics based methods and Scalable methods.

### Machine learning based methods

The authors of *Kusuma et al. (2016)* designed intelligent k-means based on spark. Intelligent k-means is a fully unsupervised learning that cluster data without any information regarding the number of clusters. A parallel implementation of biclustering using map-reduce over Spark platform was proposed by *Sarazin, Lebbah & Azzag (2014)*. For improving the selection process of k-means, (*Gao & Zhang, 2017*) combines Particle Swarm Optimization and Cuckoo-search to initiate better cluster centroid selections using spark framework. The work in *Thakur & Dharavath (2018)* proposes a hybrid approach that integrate k-means and decision tree to cluster and detect anomaly in big data. At first, k-means is applied on the data to produce the clusters and then decision tree algorithm is applied on each cluster to classify normal and anomaly instances.

In *Lighari & Hussain (2017)* the author combines rule based and k-means algorithm for the detection of network anomalies using apache spark. Rule based is used for the detection of known attacked, while k-means is used as unsupervised learning for the detection of new unknown attacks. Kdd cup dataset was used to evaluate the algorithm and 93% accuracy was achieved. A paralleled algorithm for the evolving clustering method was proposed by *Kamaruddin, Ravi & Mayank (0000)*. EMC is an online method which process one data sample on a single pass and there is no iteration required to process the same data again. These features make the algorithm highly efficient for processing contemporary real time applications where data arrive in a stream with high dimensionality. The authors evaluated the proposed algorithm using massive credit card fraud dataset and the results show its superiority over the traditional single EMC method.

### Fuzzy based methods

The authors of *Wu et al. (2017)* proposed a parallel implementation of fuzzy consensus clustering for on the Spark platform for processing large scale heterogenous data. The authors of *Win et al. (2019a)* developed a crime pattern-discovery system based on fuzzy clustering under Spark. The method uses L2 norm rather than Euclidian distance to optimize the distance computations. In another paper, the fuzzy clustering method is used under Spark to detect potential criminal patterns in large-scale spatiotemporal datasets (*Win et al., 2019b*). In *Liu et al. (2019)* the authors developed a parallel Fuzzy based image segmentation algorithm for handling big data in the agriculture field. At first, the images were converted to RGB and distributed to the available nodes in cloud. Then, the membership of pixel points to different cluster centroids were calculated. Finally, the centroids of the clusters are updated iteratively until an optimal solution is obtained. The performance of the algorithm was evaluated using the Spark platform and a significant reduction in execution time compared to Hadoop-based approach. The authors of *Bharill, Tiwari & Malviya (0000)* proposed an algorithm of fuzzy c-Means. The proposed algorithm is a modification of the Scalable Random Sampling with Iterative Optimization (SRSIO-FCM). The highlighted characteristics of this research were the elimination of the need for maintaining the membership matrix, which proved pivotal in reducing execution time.

### Statistics based methods

The authors of *Shah (2016)* used Apache Spark to perform text clustering. Two algorithms were used: k-means and LDA. LDA is Widely used technique for clustering high dimensional text data and it produces considerably higher clustering accuracy than conventional k-means. In *Lavanya, Sairabanu & Jain (2019)* the authors used gaussian mixture model on spark MLlib to cluster the zika virus epidemic. The produced clusters were useful to visualize the spread of the virus during the epidemic. The authors of *Pang et al. (0000)* implemented GMM clustering method under the framework of Spark. Gibbs sampling method is used instead of Expectation Maximization algorithm to estimate the parameters of the model. The efficiency of the algorithms was verified via multi-method comparison. The authors in *Chakravorty et al. (2014)* presented a novel distributed gaussian based clustering algorithm for analysing the behaviour of households in terms of energy consumption. Various factors such as weather conditions, type of day and time of the day were considered. The proposed algorithm under Spark shows a higher accuracy than other standard regression methods for energy consumption forecasting.

### Scalable methods

A parallel implementation of k means algorithm over spark is proposed in *Wang et al. (2016)*. The proposed algorithm involves three strategies for seeding: (1) a subset of data is selected randomly for partitioning. (2) sequentially selecting k instance based on probability. (3) stochastically selecting seeds in parallel. The efficiency of the proposed algorithm was demonstrated via experiments on large scale text and UCI datasets. In another paper, the authors addressed the issue of pre-determining the number of input clusters which is a present problem in most K-means methods by automating the number of input clusters which resulted in better clustering quality when processing large scale data (*Sinha & Jana, 2016*).

In *Backhoff & Ntoutsi (2016)*, the authors presented a scalable k-means algorithm based on spark streaming for processing real time- data. The algorithm consists of two parts. One part runs an online algorithm over the stream data and obtains only statistically relevant information and another part that uses an offline algorithm on the results of the former to produce the actual clusters. Since the algorithm only retains statistically relevant information, it's possible to save more physical spaces. In addition, the proposed algorithm can explain the evolution of the data as all the needed information is retrievable from the stored statistical information.

The authors of *Ding et al. (2017)* used k-means under Spark to cluster students' behaviors into different categories using information gathered from universities' information system management. It is a powerful technique for performing simultaneous clustering of rows and Columns in a matrix data format. This method is used extensively for the study of genes expression. The authors of *Sharma, Shokeen & Mathur (2016)* clustered satellite images in an astronomy study using in k-means++ under the spark framework. In this paper, the authors simultaneously apply k-means multiple times with different initial centroids and value of k under each iteration. The optimal value of $k$ is determined by clusters validity index for all the executions. The work in *Ben HajKacem, Ben N'Cir &*

*Essoussi (2017)* presented a Spark-based k-prototypes (SKP) clustering method for mixed large-scale data analysis. The authors exploit the in-memory operations of Spark to reduce the consumption time of MRKP method. The method was evaluated using simulated and real datasets under Spark and Hadoop platform and the results show that higher efficiency and scalability is achieved under Spark.

The authors of *Ben HajKacem, Ben N'cir & Essoussi (0000)* implemented a Scalable Random Sampling for K-Prototypes Using Spark. The algorithm randomly selects a small group of data points and approximate the cluster centers from these data. As a result, the method perform computation for only small portion of the whole data set, which result in a significant speedup of existing k-prototypes methods. A Parallel Overlapping k-means algorithm (POKM) is proposed in *Zayani, Ben N'Cir & Essoussi (2016)*. This algorithm can perform parallel clustering processes leading to non-disjoint partitioning of data. An implementation of parallel k-means with triangle inequality based on spark is proposed in *Chitrakar & Petrovic (2018)*. The method is an improved version of k-means, which is supposed to speed up the process of analysis by skipping many point-centre distance computations, which can be beneficial when clustering high dimensional data. The authors of *Fatta & Al Ghamdi (2019)* implemented k-means with triangle inequality to reduce search time and avoid redundant computation. The authors point out that the efficiency of k-means can be improved significantly using triangle inequality optimisations. A distributed possibilistic c-means algorithm is proposed in *Zhang et al. (2019)*. Possibilistic c means differ from other k-means techniques by assigning probabilistic membership values in each cluster for every input point rather than assigning a point to a single cluster.

The authors of *Solaimani et al. (2014)* implemented an adaptive k-mean using Spark stream framework for real time-anomaly detection in clouds virtual machines. The authors evaluated the method under Spark and Storm in terms of the average delays of tuples during clustering and prediction and the results indicate that Spark is significantly faster than Storm. *Mallios et al. (0000)* designed a framework for clustering and classification of big data. The framework integrates k-means and decision tree learning (ID3) algorithms. the authors evaluated the framework under Spark in cluster of 37 nodes. the results show that the proposed algorithms outperform spark machine learning library but is slightly slower than the approximate k-means.

## Hierarchical clustering

This clustering technique is composed of two approaches: agglomerative and divisive. The first approach considers every data point as a starter in its singleton cluster and the two nearest clusters are combined in each iteration until the two different points belong to a similar cluster. However, the second approach performs recursive top-down splitting. The existing hierarchical clustering methods can be divided into three subcategories: Data Mining based methods, Machine Learning based methods and Scalable methods.

### Data mining based methods

A weighted agglomerative hierarchical clustering algorithm is introduced in *Guo, Zhang & Zhang (2016)*. The algorithm was developed to analyse residents' activities in China.

The data was based on the mobile phone's connection with the nearest stations, and within a week that data was collected and stored in Spark for analysing. At first, hot areas where there are large population were identified, followed by an analysis of pedestrian's flow for each hot area. Meaningful information was obtained at less cost and higher accuracy than the traditional method of investigation. The work in *Ianni et al. (2020)* proposed a distributed-hierarchical based clustering algorithm that combines the features of the divisive and agglomerative methods. The method consists of two operations. The first operation performs a division on the domain of the dataset using the definition of binary space partition, which yields a set of coarse clusters that are then refined by identifying outliers and assigning remaining points to nearest cluster. The second operation involves an agglomerative procedure over the previously refined clusters. In *Lee & Kim (0000)*, they proposed a Distributed-based Hierarchical Clustering System for Large-Scale Semiconductor Wafers (DHCSSW) by applying the big data Spark framework to existing hierarchical clustering algorithm.

### Machine learning based methods

In *Sarazin, Azzag & Lebbah (2014)*, the authors designed clustering algorithms that can be used in MapReduce using Spark platform. Particularly, they focus on the practical and popular serial Self-organizing Map clustering algorithm (SOM). *Malondkar et al. (2019)* proposed an algorithm called Spark-GHSOM, which scales to large real-world datasets on a distributed cluster. Moreover, it also proposed a new distance hierarchy approach for mixed attribute datasets and generated a multi-level hierarchy of SOM layers.

### Scalable methods

In *Jin et al. (2015)*, a parallel algorithm of Single-linkage Hierarchical Clustering was proposed by formulating the problem as a Minimum Spanning Tree problem. The algorithm was evaluated using two large datasets with different distributions. The authors observed that spark is totally successful for the parallelization of linkage hierarchical clustering with acceptable scalability and high performance. The work in *Solaimani et al. (0000)* proposed a system to detect anomaly for multi-source VMware-based cloud data center. The framework monitors VMware performance stream data (e.g., CPU load, memory usage, etc.) continuously. The authors of *Hassani et al. (2016)* presented an incremental hierarchical density-based stream clustering algorithm based on cluster stability.

## Density-based clustering

Density-based clustering approaches in comparison with other types of clustering algorithms have some superiorities, such as clustering arbitrary shape groups of data regardless of the geometry and distribution of data, robustness to outliers, independence from the initial start point of the algorithm, and its deterministic and consistent results in the repeat of the similar algorithm. Motivated by these features, several studies have been conducted on the parallelization of Density clustering method over Spark. Density-based clustering methods can be divided into four subcategories: Graph based methods, Data Mining based methods, Machine Learning based methods and Scalable methods.

### Graph based methods

In *Rui et al. (2017)*, the authors proposed a parallel implementation of density peaks clustering algorithm based on Spark's GraphX. The method was evaluated using spark and the results indicate that spark can perform up to 10x time faster compared to Hadoop map-reduce implementation. *Zhou & Wang (0000)* proposed a distributed parallel algorithm of structure similarity clustering based on Spark (SparkSCAN) to cluster directed graph. Similarly, the authors of *Kim et al. (2018)* exploited the advantage of the in-memory computation feature of spark to design a distributed network algorithm called CASS for clustering large-scale network based on structure similarity. Optimization approaches such as Bloom filter and shuffle selection are used to reduce memory usage and execution time. *Lulli, Dell'Amico & Ricci (2016)* designed a distributed algorithm that produces an approximate solution to the exact DBSCAN clustering. The method uses vertex-centric instead of Euclidean distance whereby a neighbourhood graph is computed.

### Data mining based methods

In another paper *Han et al. (2018a)*, the authors presented a fast parallel DBSCAN algorithm using Spark to get around the shuffle operation. Each executor computes the partial clusters locally. The merging process is deferred until all the partial clusters have been sent back to the driver. In *Hosseini & Kourosh (2019)* the authors propose a scalable distributed density based hesitant fuzzy clustering for finding similar expression between distinct genes. The proposed method benefits from the robustness of density-based clustering against outliers and from the weighted correlation operators of hesitant fuzzy clustering to measure similarity. The output clusters are based on the content of the neighbour graph. *Aryal & Wang (2018)* designed and implemented a scalable Shared Nearest Neighbours clustering called SparkSNN over spark framework. Shared Nearest Neighbours is proven efficient for handling high-dimensional spatiotemporal data. The algorithm was evaluated in terms of scalability and speed-up using Marylanf crime data, the results demonstrated the effectiveness of the proposed algorithm.

### Machine learning based methods

An algorithm based on adaptive density estimation is proposed for distributed big data approach and tested on some prevalent datasets. This algorithm has no dependency, and every step of the algorithm executes independently. Bayesian Locality Sensitive Hashing (LSH) is used to divide the input data into partitions. The outliers are filtered out by locality preservation, which makes this approach robust. The clusters are made very much homogenous via density definition on Ordered Weighted Averaging distance *Hosseini & Kiani (2018)*. A scalable distributed density-based clustering for performing multi-regression tasks is proposed in *Corizzo et al. (2019)*. In this work, locality sensitive hashing is used to enable the algorithm to handle high dimensional data. A distributed clustering algorithm named REMOLD is introduced in *Liang et al. (2017)*. A two-step strategy has been applied in the REMOLD algorithm. In the first step, it uses the LSH partitioning method for balancing the effect of runtime and local clustering while in the second step the partitions are clustered locally and independently using Kernel-density and Higher-density

nearest neighbour. Gaussian distribution is used to model the local clusters. These models are eventually assembled at a central server to form the global clusters.

### Scalable methods

In *Luo et al. (2016)*, a parallel implementation of DBSCAN algorithm (S_ DBSCAN) based on spark is proposed. The algorithm is divided into three stages; partitioning the input data based on random sampling; perform local DBSCAN in parallel to generate partial clusters; merge the partial clusters based on the centroid. The algorithm can quickly realize the mergers and divisions of clustering results from the original data. The authors compared the performance of their parallel algorithm with a serial version on the Spark platform for massive data processing and an improvement in performance was demonstrated. *Han et al. (2018b)* proposed a scalable parallel implementation of DBSCAN algorithm in Apache spark by applying a partitioning strategy. The algorithm uses a kd-tree in order to reduce the search time. To achieve better performance and scalability, a partitioning technique is applied to produce balanced sub-domains, which can be computed within Spark executors. An implementation of DBSCAN algorithm using spark is proposed in *Baralis, Garza & Pastor (2018)*. Initially, a pre-processing step is applied on the dataset to produce a set of representative points while retaining the original data distribution and density information. This enables the algorithm to scale up to large scale data. The new set is then used as an input to the algorithm for clustering. A real-time density-based clustering algorithm (RT-DBSCAN) is proposed in *Gong, Sinnott & Rimba (0000)*. RT-DBSCAN is an extension of dbscan for supporting streamed data analysis. The algorithm employs the concept of spatiotemporal distance for clustering spatio-temporal data. The algorithm was implemented over spark stream and evaluated using social media content.

## Clustering optimization

Some Spark-based clustering techniques, especially the k-means based methods, were supported by optimization techniques to improve their clustering results. Due to the rise of AI based computing in recent years, some research works have utilized AI tool in enhancing the clustering methods while leveraging the benefits of Big Data platforms such as Spark. Other studies adapt optimization techniques to improve the performance of clustering methods. *Sherar & Zulkernine (2017)* proposed a hybrid method composed of PSO and k-means using apache spark. The diversity of the swarm ensures that a global search is conducted, hence, the resulting cluster centroids are not dependent on the initial choice. The approach was compared with stand-alone k-means and it showed better performance in terms of convergence. *Hasan et al. (2019)* proposed an adaptive swarm-based clustering for stream processing of twitter data. Initially, fuzzy c-means is applied as pre-processing step to produce the initial cluster centres, then the clusters are further optimized using adaptive particle swarm optimization.

The authors of *Wang & Qian (2018)* and (*Bonab et al., 2015*) combined the robust artificial bee colony algorithm with the powerful Spark framework for large scale data analysis. The characteristics of ABC makes the algorithms avoid local minimum while Spark in memory computation accelerates the speed of computation and convergence

time. The KDD CUP 99 data was utilized to verify the effectiveness of the method. The experimental results show that the algorithm produce high clustering quality and nearly as fast as the serial algorithms. Other unsupervised learning such as self-organised map has also been proposed (*Sarazin, Azzag & Lebbah, 2014*). To tackle high dimensional data, subspace clustering was proposed by *Sembiring, Jasni & Embong (2010)*.

## DISCUSSION AND FUTURE DIRECTION

From the discussion of the previous section, we note that most existing methods (see Table 1) have addressed the *volume* characteristic of the Big Data used in their experiments. However, few existing methods have shown that their methods support the *variety* and *velocity* characteristics of the used Big Data. Additionally, most methods used *real* Big Data validate their proposed methods as seen in Table 1. From Table 1, we conclude that there is a lot of room for research in clustering methods to support the characteristics of variety and velocity of Big data since only few works have addressed these issues.

A fundamental assumption of most clustering algorithms is that all data features are considered equally important. However, such approach often fails in high dimensional space. A subspace clustering overcome the issue of high dimensional data by establishing a set of features that it supposes to be most significant for each cluster.

Since the Big data platforms were only developed in the last few years, the existing clustering problems adapted to such platforms were extensions of the traditional clustering techniques. Researchers are yet to develop clustering techniques that are native to the Big Data platforms such as Spark. The research direction of adapting the optimization techniques such as PSA, Bee colony and ABC to smoothly work with Spark is yet to be investigated by researchers who are interested in clustering Big Data. Another area of research that is has not been fully investigated is adopting Fuzzy-based clustering algorithms on Spark.

In general, due to the infancy of Spark-based clustering algorithms, only few researchers attempted designing techniques that leverage the potential of parallelism of Spark in cluster Big Data. In the coming years, we foresee a large influx of research works in this important area of Spark-based clustering of Big Data. Particularly, there are ample opportunities in future research to utilize AI tools in clustering data while leveraging the benefits of Big Data platforms such as Spark.

In Table 2, we note that most of the papers used in this survey were extracted from the IEEE Explorer. However, the other data sources shown in Table 2 were of great benefit to this survey. An interesting finding was shown in Table 3, where most the existing Spark-based Clustering were published in the years 2016–2019. This indicates that clustering methods that leverage Big Data platforms is still in its early days and there is a lot of potential of research in this area.

In summary, we highlight three new research directions:

- Utilizing AI tools in clustering data while leveraging the benefits of Big Data platforms such as Spark.

**Table 2** Shows the data sources of the Spark-based clustering papers.

| Data source | Paper |
|---|---|
| IEEE Explorer | *Dave & Gianey (2016), Hu et al. (2018), Fasheng & Xiong (2011), Ajin & Kumar (2016), Verma, Mansuri & Jain (2016), Xin et al. (2013), Xu & Tian (2015), Manwal & Gupta (2017), Shanjiang et al. (2018), Wang et al. (2016), Sinha & Jana (2016), Kusuma et al. (2016), Backhoff & Ntoutsi (2016), Ding et al. (2017), Sarazin, Lebbah & Azzag (2014), Ben HajKacem, Ben N'Cir & Essoussi (2017), Wu et al. (2017), Zayani, Ben N'Cir & Essoussi (2016), Chitrakar & Petrovic (2018), Win et al. (2019b), Liu et al. (2019), Solaimani et al. (2014), Chakravorty et al. (2014), Lighari & Hussain (2017), Jin et al. (2015), Guo, Zhang & Zhang (2016), Solaimani et al. (0000), Lee & Kim (2018), Hassani et al. (2016), Sarazin, Azzag & Lebbah (2014), Luo et al. (2016), Han et al. (2018b), Aryal & Wang (2018), Liang et al. (2017), Sherar & Zulkernine (2017), Wang & Qian (2018), Sarazin, Azzag & Lebbah (2014b)* |
| Elsevier | *Bhadani & Jothimani (2016), Ben HajKacem, Ben N'Cir & Essoussi (2017), Rui et al. (2017)* |
| Springer | *Othman et al. (2018), Baltas, Kanavos & Tsakalidis (0000), Zerhari, Lahcen & Mouline (2015), Rujal & Dabhi (2016), Sood, Akshay & Singh (2019), Ketu & Agarwal (2015), Santhi & Jose (2018), Jain (2010), Huang (1998), Ben HajKacem, Ben N'cir & Essoussi (0000), Win et al. (2019a), Gao & Zhang (2017), Pang et al. (0000), Mallios et al. (0000), Thakur & Dharavath (2018), Kamaruddin, Ravi & Mayank (0000), Han et al. (2018a), Corizzo et al. (2019), Zhou & Wang (0000), Gong, Sinnott & Rimba (0000), Bonab et al. (2015)* |
| Google Scholar | *Kim (2018), Sharma, Shokeen & Mathur (2016), Gousios (2018), Aziz, Zaidouni & Bellafkih (0000), Mishra, Pathan & Murthy (2018), Jiang et al. (2010), Shobanadevi & Maragatham (2017), Lavanya, Sairabanu & Jain (2019), Kim et al. (2018), Lulli, Dell'Amico & Ricci (2016), Hasan et al. (2019)* |
| ResearchGate | *Shoro & Soomro (2015), Salloum et al. (2016), Assefi et al. (2017), Armbrust et al. (2015), Hosseini & Kiani (2018), Hartigan, Wong & Algorithm (1979), Shah (2016), Fatta & Al Ghamdi (2019), Sarazin, Azzag & Lebbah (2014b)* |
| Science Direct | *Labrinidis & Jagadish (2012), Shirkhorshidi et al. (0000), Rotsnarani & Mrutyunjaya (2015), Huang et al. (2017), Zhang et al. (2019), Maheshwar & Haritha (2016), Ianni et al. (2020), Malondkar et al. (2019), Rui et al. (2017), Hosseini & Kourosh (2019)* |

- Clustering methods to support the characteristics of variety and velocity of Big data. Additionally, support new aspects of clustering such as concept drift, scalability, integration, fault-tolerance, consistency, timeliness, load balancing, privacy, and incompleteness, etc.
- Clustering methods to utilize Spark as it is an efficient Big Data platform.

**Table 3  Shows which papers in the survey were published in each of the last 6 years.**

| Year of publication | Papers | Number of papers |
| --- | --- | --- |
| 2014 | *Wang et al. (2016), Santhi & Jose (2018), Sarazin, Lebbah & Azzag (2014), Solaimani et al. (2014), Chakravorty et al. (2014), Solaimani et al. (0000), Sarazin, Azzag & Lebbah (2014b)* | 8 |
| 2015 | *Verma, Mansuri & Jain (2016), Shoro & Soomro (2015), Bhadani & Jothimani (2016), Rotsnarani & Mrutyunjaya (2015), Shobanadevi & Maragatham (2017), Jin et al. (2015), Kim (2018), Bonab et al. (2015)* | 8 |
| 2016 | *Bhadani & Jothimani (2016), Dave & Gianey (2016), Ajin & Kumar (2016), Baltas, Kanavos & Tsakalidis (0000), Gousios (2018), Assefi et al. (2017), Maheshwar & Haritha (2016), Wang et al. (2016), Sinha & Jana (2016), Kusuma et al. (2016), Backhoff & Ntoutsi (2016), Sharma, Shokeen & Mathur (2016), Zayani, Ben N'Cir & Essoussi (2016), Shah (2016), Bharill, Tiwari & Malviya (0000), Pang et al. (0000), Mallios et al. (0000), Guo, Zhang & Zhang (2016), Hassani et al. (2016), Zhou & Wang (0000), Lulli, Dell'Amico & Ricci (2016)* | 21 |
| 2017 | *Baltas, Kanavos & Tsakalidis (0000), Salloum et al. (2016), Armbrust et al. (2015), Xu & Tian (2015), Shirkhorshidi et al. (0000), Rujal & Dabhi (2016), Ding et al. (2017), Ben HajKacem, Ben N'Cir & Essoussi (2017), Wu et al. (2017), Gao & Zhang (2017), Lighari & Hussain (2017), Kamaruddin, Ravi & Mayank (0000), Rui et al. (2017), Liang et al. (2017), Sherar & Zulkernine (2017)* | 15 |
| 2018 | *Othman et al. (2018), Hu et al. (2018), Kim (2018), Xin et al. (2013), Zerhari, Lahcen & Mouline (2015), Sood, Akshay & Singh (2019), Jiang et al. (2010), Shanjiang et al. (2018), Huang et al. (2017), Ketu & Agarwal (2015), Ben HajKacem, Ben N'cir & Essoussi (0000), Chitrakar & Petrovic (2018), Thakur & Dharavath (2018), Lee & Kim (2018), Luo et al. (2016), Han et al. (2018b), Han et al. (2018a), Kim et al. (2018), Baralis, Garza & Pastor (2018), Aryal & Wang (2018), Gong, Sinnott & Rimba (0000), Wang & Qian (2018)* | 22 |
| 2019 | *Aziz, Zaidouni & Bellafkih (0000), Win et al. (2019a), Fatta & Al Ghamdi (2019), Zhang et al. (2019), Win et al. (2019b), Liu et al. (2019), Lavanya, Sairabanu & Jain (2019), Hosseini & Kiani (2018), Corizzo et al. (2019), Hosseini & Kourosh (2019), Hasan et al. (2019), Sembiring, Jasni & Embong (2010)* | 12 |
| 2020 | *Ianni et al. (2020)* | 1 |

# CONCLUSIONS

As a consequence of the spread of smart devices and appearance of new technologies such as IoT, huge data have been produced on daily bases. As a result, the concept of Big Data has appeared. Unlike the traditional clustering approaches, Big Data clustering requires advanced parallel computing for better handling of data because of the enormous volume

and complexity. Therefore, this work contributes to the research in this area by providing a comprehensive overview of existing Spark-based clustering techniques on Big data and outlines some future directions in this area.

Due to the infancy of the Big data platforms such as Spark, the existing clustering techniques that are based on Spark are only extensions of the traditional clustering techniques. There is still big room for developing clustering techniques designed specifically for Spark making use of the random distribution of data onto Spark partitions, called RDDs, and the parallel computation of data in the individual RDDs. Through this survey we found that most existing Spark-based clustering method support the volume characteristic of Big Data ignoring other characteristics. Therefore, future research should focus on other characteristics as well such as variety and velocity. Additionally, future Spark-based clustering method should investigate new features such as concept drift, scalability, integration, fault-tolerance, consistency, timeliness, load balancing, privacy, etc.

### Funding
The authors received support from the Deanship of Scientific Research at Prince Sattam Bin Abdulaziz University for this research. The funders had no role in study design, data collection and analysis, decision to publish, or preparation of the manuscript.

### Grant Disclosures
The following grant information was disclosed by the authors:
Deanship of Scientific Research at Prince Sattam Bin Abdulaziz University.

### Competing Interests
The authors declare there are no competing interests.

### Author Contributions
- Mozamel M Saeed conceived and designed the experiments, prepared figures and/or tables, and approved the final draft.
- Zaher Al Aghbari conceived and designed the experiments, analyzed the data, prepared figures and/or tables, authored or reviewed drafts of the paper, and approved the final draft.
- Mohammed Alsharidah performed the experiments, analyzed the data, authored or reviewed drafts of the paper, and approved the final draft.

### Data Availability
No code or raw data is involved in this research as this is a literature review.

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
