# Peer review of "Big data clustering techniques based on Spark: a literature review"

_PeerJ Computer Science, doi:10.7717/peerj-cs.321_

## Round 0.1 · original submission · Major Revisions

The reviewers raised some critical issues. The authors are suggested to address these issues through a major revision.

Reviewer 1 ·

Basic reporting

No comment.

Experimental design

No comment.

Validity of the findings

No comment.

Additional comments

This survey studies the previous literature and the existing applications on big data clustering techniques based on Spark, and the authors outline a commonly used pipeline in clustering methods based on Spark in a systematic way and focus on discussing by giving a comprehensive review and 93 direct researchers to the strong and weak points of the current Spark-based clustering methods. After that, they discuss the different Spark clustering algorithms. Finally, they present the discussion on the clustering big data using Spark and future direction.
However, the topic is quite old. This paper fails to represent a high-quality survey with in-depth analysis, comparison, and discussion. The specific comments are shown as follows.
1. As a survey paper, I don’t see many tables to make comparisons and no timeline to represent the evolution of big data clustering methods, which makes this paper quiet weak.
2. To review the data clustering based on Spark is a large topic, I recommend the authors to pick a distinguished perspective to study this topic, instead of attempting to review multiple aspects of it.
3. As we step into the AI era, I also recommend the authors could add some machine learning-based works to make this survey more consistent with the technique changes.
4. It would be better to give new insight into the challenges and future research direction.
5. The figures in this paper are not clear enough. The text in the figures are blurred.

Reviewer 2 ·

Basic reporting

Designing and developing distributed and scalable machine learning algorithms, including clustering algorithms, is an active research topic. Apache Spark is one of the distributed general-purpose cluster-computing frameworks popular among both industry and academia. Evaluation of the developments of clustering algorithms based on Spark technology is timely and useful. A few good reviews around clustering algorithms in the big data paradigm is already available. However, a proper evaluation of distributed clustering algorithms primarily focusing on Spark and its characteristics beneficial.

Experimental design

The authors have followed a scientific approach to identify relevant literature. The review has been broadly organized into three main categories based on the proposed taxonomy. The authors have identified a fair number of recent publications related to the scope. However, it would have been easy to understand if different algorithms were discussed under a suitable sub-heading under each branch.

Validity of the findings

The authors have briefly discussed the characteristics of each algorithm they have reviewed. However, it is recommended to include a systematic approach to evaluate each algorithm to show the reviewed algorithms' strengths and weaknesses. Further, you may include each algorithm's ability to tackle key big data challenges - volume, variety and velocity. Additionally, specific criteria can be proposed to validate the algorithms' relative strengths and weaknesses through experiments.

The proposed taxonomy is well known and can be applied for this scope as well. Fig 3 can be improved by mentioning each algorithm's name under each branch, so that the readers can easily capture the overview of the reviewed algorithms immediately.

Additional comments

In general, the flow of the paper is good. However, the readability of the section V may be increased by introducing appropriate sub-headings.

Reviewer 3 ·

Basic reporting

1. The English Language should be improved to avoid typographical errors. Some examples of typographical errors include lines 69, 143, 181, 232, 257, 325, 340, 447, 470 and 539.

2. The introduction is detailed and well referenced.

3. The structure of the paper needs improvement. For instance, the abstract did not provide a summary of what the systematic review is all about. There was too many background information in the abstract. The abstract is expected to give a little background briefly, then the problem, methods, results, and significance. Besides, I think the background section should come before the literature review section. Moreover, the discussion section should deliberate on the findings/results, but the authors used this section for another literature review or related work.

4. Although there are existing literature or systematic reviews on big data analytics, the authors dwelt specifically on clustering analysis of big data with Apache Spark, which is commendable.

5. The subject matter was adequately introduced.

Experimental design

1. The paper is a systematic literature review and has applicability in other fields, most especially where they deal with a large scale of data. It is within the scope of the journal.

2. The background section needs improvement, especially under the challenges of clustering big data. The authors looked at one direction, the V’s. It will be more comprehensive if authors could also look at other challenges like concept drift, scalability, integration, fault-tolerance, consistency, timeliness, load balancing, privacy, and incompleteness, etc.

Validity of the findings

1. While the paper has tried to address big data clustering with Apache spark, some things are still missing. The authors claimed to address five questions, but only the first two were answered. One would have expected the authors to explicitly pick the questions one after the other and provided answers to them from the selected 91 papers. Even the first two that were addressed were implied.

2. The conclusion still needs improvement. The conclusion is not linked with how the five questions raised in the paper were answered. The conclusion slightly mentioned future direction but still needs more elaboration.

Additional comments

1. A summary of the final 91 selected papers should be presented in a table. This should reflect the number of papers finally selected from each of the sources (database) used.

2. Line 393-396, the definitions of divisive and agglomerative hierarchical clustering were interchanged.

3. Referencing style is not consistent. Some started with Authors’ surname while others with initials.

4. Line 156-161. A better spark architecture should replace figure 1. The components mentioned, driver, executors, and RDD are not reflected in the figure.

5. The research question in line 232 is not a complete statement. I think the authors missed “are the”.

6. In line 257, the correct the spelling, “Elsevier”

7. All the three figures in the paper were not cited.

---

## Round 0.2 · Minor Revisions

This manuscript has been well improved. However, there are still some minor issues to be addressed.

Reviewer 1 ·

Basic reporting

The paper has been revised according to the comments. However, it still needs to be further improved.

Experimental design

No comment.

Validity of the findings

No comment.

Additional comments

The paper has been revised after a first review process. It has been improved in most parts but some shortcomings are still present.
1. The identifier of chapters is not sequential.
2. The writing style of the manuscript is inconsistent. For example, some subtitles are capitalized, others are not.
3. The terms used in this manuscript is also inconsistent, for example, “k means”, “k-means” , “k-mean”.

Reviewer 3 ·

Basic reporting

1. There is an improvement in the use of language as compared to the previous version of the paper. However, a Langauge editor can still work on the paper for further improvement. For instance, in line 302, there should be space between 'methods' and 'The'. Line 525, correct the spelling of 'volumn'

2. Authors have improved on the structure but there are still some things to look into. Authors should look at my comments under the validity of findings

Experimental design

Survey Findings/Results should come immediately after the Survey Methodology. Titling section V with "Spark Clustering Algorithm" (the section where the authors claim that it provides most of the answers to the survey questions) does not connote to the readers as the answers to the survey questions.

Validity of the findings

All the research findings, that is, answers to the five survey questions should be under a section titled 'Survey Findings' or simply put, 'Results'. Answers to the questions should not be implied as it is presently. Authors should pick them one after the other and address them.

For example, authors may decide to use the following format:

V. Survey Findings
The findings of the survey in line with the survey questions are presented in this section.

Research Question 1: What are the types of Spark-based clustering methods?
Authors will now provide answers from the selected 91 papers and answer the question.

Research Question 2:
.
.
.
Research Question 5

Following this format will make it clear to the audience.

Additional comments

1. Line 238 - Authors may remove the statement, "These questions ... paper". As I have suggested earlier, all the answers to the survey questions should appear under one section.

2. Section V is another literature review. Research findings should be relevant answers to the survey questions from the selected 91 papers and not another literature review.

3. The same observation is noted in the discussion section. Authors should focus on the implications of the survey findings/results and not another literature review.

---

## Round 0.3 · accepted · Accept

All reviewers' comments have been well addressed. I appreciate all the efforts the authors have made in improving the quality of the paper.